# Sensor-as-a-Service: Convergence of Sensor Analytic Point Solutions (SNAPS) and Pay-A-Penny-Per-Use (PAPPU) Paradigm as a Catalyst for Democratization of Healthcare in Underserved Communities

**DOI:** 10.3390/diagnostics10010022

**Published:** 2020-01-01

**Authors:** Victoria Morgan, Lisseth Casso-Hartmann, David Bahamon-Pinzon, Kelli McCourt, Robert G. Hjort, Sahar Bahramzadeh, Irene Velez-Torres, Eric McLamore, Carmen Gomes, Evangelyn C. Alocilja, Nirajan Bhusal, Sunaina Shrestha, Nisha Pote, Ruben Kenny Briceno, Shoumen Palit Austin Datta, Diana C. Vanegas

**Affiliations:** 1Agricultural and Biological Engineering, Institute of Food and Agricultural Sciences, University of Florida, Gainesville, FL 32611, USA; tvlmorgan@ufl.edu (V.M.); emclamore@ufl.edu (E.M.); shoumen@mit.edu (S.P.A.D.); 2Natural Resources and Environmental Engineering, Universidad del Valle, Cali 760026, Colombia; lisseth.casso@correounivalle.edu.co (L.C.-H.); irene.velez@correounivalle.edu.co (I.V.-T.); 3Interdisciplinary Group for Biotechnological Innovation and Ecosocial Change BioNovo, Universidad del Valle, Cali 760026, Colombia; 4Biosystems Engineering, Department of Environmental Engineering and Earth Sciences, Clemson University, Clemson, SC 29631, USA; dbahamo@g.clemson.edu (D.B.-P.); mccourt923@aol.com (K.M.); 5Mechanical Engineering, Iowa State University, Ames, IA 50011, USA; ghjort@iastate.edu (R.G.H.); carmen@iastate.edu (C.G.); 6School of Computer Engineering, Azad University, Science and Research Branch, Saveh 11369, Iran; sahar.bahramzade@gmail.com; 7Global Alliance for Rapid Diagnostics, Michigan State University, East Lansing, MI 48824, USA; alocilja@msu.edu (E.C.A.); nirajanbhusal@gmail.com (N.B.); 8Biosystems and Agricultural Engineering, Michigan State University, East Lansing, MI 48824, USA; 9School of Medical Sciences, Kathmandu University, Kathmandu 44600, Nepal; 10Dhulikhel Hospital, Kathmandu University, Kavrepalanchok 45200, Nepal; stha.sunaina@gmail.com (S.S.); potenisha@gmail.com (N.P.); 11Instituto de Investigacion en Ciencia y Tecnologia, Universidad Cesar Vallejo, Trujillo 13100, Peru; rubskenny@gmail.com; 12Hospital Victor Lazarte Echegaray, Trujillo 13100, Peru; 13Institute for Global Health, Michigan State University, East Lansing, MI 48824, USA; 14MIT Auto-ID Labs, Department of Mechanical Engineering, Massachusetts Institute of Technology, 77 Massachusetts Avenue, Cambridge, MA 02139, USA; 15MDPnP Interoperability and Cybersecurity Labs, Biomedical Engineering Program, Department of Anesthesiology, Massachusetts General Hospital, Harvard Medical School, 65 Landsdowne Street, Cambridge, MA 02139, USA; 16NSF Center for Robots and Sensors for Human Well-Being, Purdue University, 156 Knoy Hall, Purdue Polytechnic, West Lafayette, IN 47907, USA

**Keywords:** sensor analytic point solutions (SNAPS), environmental health, poverty, pay-a-penny-per-use (PAPPU), public health

## Abstract

In this manuscript, we discuss relevant socioeconomic factors for developing and implementing sensor analytic point solutions (SNAPS) as point-of-care tools to serve impoverished communities. The distinct economic, environmental, cultural, and ethical paradigms that affect economically disadvantaged users add complexity to the process of technology development and deployment beyond the science and engineering issues. We begin by contextualizing the environmental burden of disease in select low-income regions around the world, including environmental hazards at work, home, and the broader community environment, where SNAPS may be helpful in the prevention and mitigation of human exposure to harmful biological vectors and chemical agents. We offer examples of SNAPS designed for economically disadvantaged users, specifically for supporting decision-making in cases of tuberculosis (TB) infection and mercury exposure. We follow-up by discussing the economic challenges that are involved in the phased implementation of diagnostic tools in low-income markets and describe a micropayment-based systems-as-a-service approach (pay-a-penny-per-use—PAPPU), which may be catalytic for the adoption of low-end, low-margin, low-research, and the development SNAPS. Finally, we provide some insights into the social and ethical considerations for the assimilation of SNAPS to improve health outcomes in marginalized communities.

## 1. Environmental Burden of Disease

According to the World Health Organization (WHO), environmental factors including unsafe water, poor sanitation, air pollution, and unintentional exposure to hazardous chemical and biological agents are root causes for the burden of disease, disability, and death in the developing world [1,2] Impoverished communities living in polluted and crowded environments are much more susceptible to the double burden of infective and non-communicable diseases, and this situation is often compounded by a lack of adequate infrastructure, weak environmental policy, and deficient or inequitable healthcare systems that disfavor economically challenged users [3,4,5,6,7,8]. Despite the global efforts to reduce poverty, indicators of health disparities between disadvantaged and affluent populations continue to persist. For instance, the 2018 World Bank estimates showed that, on average, there is a 12-fold difference in the mortality rate of infants between low- and high-income populations [9], but in countries experiencing extreme deprivation such as Somalia and Sierra Leone, this rate is nearly 20-fold higher than the average rate in wealthy nations. In 2016, diarrheal diseases linked to poor sanitation and the consumption of contaminated food and water were responsible for 1.6 million deaths, 90% of which occurred in South Asia and sub-Saharan Africa [10,11] The per capita burden of disease from inhalation exposure to airborne polycyclic aromatic hydrocarbons (by-products of fuel combustion) has been found to be nearly 33-fold higher in India compared to the USA [12,13].

Nonetheless, it is important to note that due to the myriad ways in which socioeconomic and environmental factors interact, it is very difficult to establish highly detailed associations of single environmental risk factors with epidemiological outcomes [14,15,16,17]. Moreover, environmental factors rarely occur in isolation; for example, a population can be exposed to a combination of pollutants from different sources, which could result in additive or synergistic effects and symptoms, making medical diagnostic processes extremely cumbersome [1]. In addition to limited access to healthcare systems, the problem is compounded by the relatively high cost of clinical testing, which may cause many illnesses to go under-reported or mis-diagnosed in economically challenged populations [18]. Despite the complexities involved in linking environmental and socioeconomic factors to epidemiological outcomes, there is no question that such factors can result in serious public health problems, particularly in low-income communities that bear the largest proportion of the burden of environmentally-related diseases [19,20].

Undoubtedly, much of the economic strain from both infectious and non-communicable diseases associated with unhealthy environments could be effectively diminished through preventive strategies that tackle associated risk factors [18,21]. One promising approach for addressing health risk factors in low-income communities is the deployment of integrated technologies for data-informed decision support such as sensor analytic point solutions (SNAPS). The concept of SNAPS was recently introduced as part of a platform approach to converge sensor data and analytics to deliver data-informed decision support for a number of applications, including healthcare [22]. Even though thousands of sensors and point-of-care diagnostic tools have been developed in research labs around the world in the past few decades, the large majority of these technologies have not yet translated into implementable solutions due to different obstacles including the unsuitability of operation under real-world conditions, high fabrication and operation costs (which limits market penetration and profitability), and a lack of convergence with other technologies to yield actionable information for the user [23].

Consider, for instance, the case of diarrheal diseases associated with *Escherichia coli* infection from the ingestion of contaminated food or water, which significantly contributes to the mortality and morbidity of children under five years of age in African and Eastern Mediterranean countries [24]. By conducting a literature search on the Web of Science, we found that, in the past 10 years, 303 research articles have been published in peer-reviewed journals that have portrayed the development of *E. coli* biosensors. However, only a small fraction of these papers has included claims such as real-sample testing (~29%), low-cost fabrication (~10%), portability (~9%), and user-friendly operation (~2%) (the complete report from this search is available in the Appendix A). 

In this manuscript, we provide examples of SNAPS that have been tested in field conditions, within the context of low-income communities. The first example was developed for assisting the early diagnosis of infectious disease and the prevention of public health outbreaks, and the second example supports decision making in cases of human exposure to an environmental pollutant. We also propose the concept of pay-a-penny-per-use (PAPPU) as a potential paradigm to reduce economic barriers to implement SNAPS in economically-deprived regions. The two examples of field-tested SNAPS are at different stages of maturity, providing insight into the design process and logic flow. Finally, we provide some insights on the social and ethical considerations for the effective use of SNAPS in assisting users and improving health outcomes in underserved communities.

## 2. Examples of SNAPS-ART

Near real-time qualitative decisions are often key for rapid response. SNAPS make up a tool that uses sensor data to provide a response at the point of use with minimal analytics. If two or more factors must be considered by the human-in-the-loop to take a decision, artificial reasoning tools (ARTs) are implemented. ARTs make up a data fusion layer that combines sensor data and displays suggestions or information on the user’s mobile device. In principle, SNAPS are designed to offer “point solutions,” which implies a rapid binary output (yes/no) based on the data captured from the sensor signal (for example, sensor binds to an analyte). However, even in rudimentary scenarios, a single source of binary data may fail to provide basic information. Hence, the need for artificial reasoning tools (ARTs), which are light-weight middleware (software that sits in the “middle”) embedded with preliminary logic to decide what is the meaning of the data and what information may be conveyed (displayed) for the end-user. By introducing a modular ART, the user takes advantage of a combinatorial variant configuration menu to change, adapt, or introduce new reasoning/logic in the middleware by re-programming the logic “buckets” by simply re-shuffling and inserting the user’s preferred choices from a repertoire of pre-programmed logic [22]. 

There are many complex layers to a system-level solution to ease the environmental burden on impoverished communities. Velez-Torres et al. [25] recently developed a circular system framework for integrating analytic tools (such as SNAPS) with social action research (Closed-loop integration of social action and analytical science research, CLISAR). The CLISAR framework is a transdisciplinary approach that involves analytical tools such as sensors for informing community action that is related to, for example, public health, environmental issues, or food security. Beyond simple commercial colorimetric detection strips that are used in development of CLISAR, information derived from SNAPS can transform this system by supporting decision-making processes that are aimed at improving the health outcomes of marginalized communities. 

Herein, we suggest a conceptual approach for selecting and implementing the type of diagnostic tools for implementation of SNAPS (see Figure 1). The examples that follow in the subsequent section used a five-step process that followed a closed-loop approach similar to CLISAR and other circular economic models [25,26]. The first step is to understand the specific problem as well as the social and economic context where decision-support technology may be needed. The next step is to identify readily available resources and then design diagnostic tools for creating a technology portfolio (sensors, analytics software, portable hardware, etc.). The third step involves the selection of the most appropriate tools to create SNAPS based on technical capabilities as well as interactive feedback from stakeholders. In step four, scientists and end-users test technology prototypes in field conditions by using established participatory methodologies. Finally, the results from the proof-of-concept testing are used to evaluate and refine the technology. This process is repeated until a solution meets user expectations and desired performance characteristics. The concept is based on principles of circular systems and convergent thinking [25,26], where technology refinement may occur by using reductionist or parallel approaches. Below, we present two examples of how this conceptual model is applied in real-world settings. The first example is in advanced stage field-testing (refinement and technology improvement, with some elements in the second circular phase), while the second example is in the early phase of development (tool selection and technology transfer).

### 2.1. Early Assessment of Tuberculosis in Vulnerable Populations

In 2017, 1.6 million people died from tuberculosis (TB) globally, and there were 10 million new TB cases that occurred in the same year [27]. TB has surpassed HIV as the leading infectious disease killer worldwide since 2014 [28]. Furthermore, multidrug-resistant and extensively drug-resistant TB (MDR/XDR-TB) are current global public health threats. The 2017 Moscow Ministerial Declaration on ending TB, involving 120 countries and over 800 partners, identified “to advance research and development of new tools to diagnose, treat and prevent TB” as one of four action items [29]. This meeting was followed in 2018 by a United Nations (UN) General Assembly first-ever high-level meeting to accelerate efforts to end TB [30].

The care of TB patients starts with accessible and affordable diagnosis. The majority of TB patients live in poor conditions and in geographically remote areas. Culture-based techniques are the gold standard for diagnosis, but this is relatively expensive and results take six-to-eight weeks [31]. For decades, TB diagnosis has relied on direct sputum smear microscopy (SSM) in many countries [31]. SSM is fast, inexpensive, facile, and specific for detecting *Mycobacterium tuberculosis* (Mtb) in high incidence areas [31,32,33,34]. SSM does not require a highly specialized apparatus and is therefore very suitable for low-resource settings [31,33]. However, the accuracy of SSM is only 25–65%, which is considerably lower than the standard culture technique, and its limit of detection is about 10,000 colony forming units per milliliter (CFU/mL) [34,35]. In a recent study involving hundreds of specimens tested with culture, SSM, and the Xpert MTB/RIF system, the SSM method exhibited an average accuracy of 54% for respiratory samples and 50% for non-respiratory samples [36]. Furthermore, the overall performance of SSM depends on different variables including the type of lesion, the type and number of specimens, the specific *Mycobacterial* species, the staining technique, and the competence of the microscopist [35]. In a 2014 survey, 22 high-burden countries conducted 78 million sputum smears valued at 137 million USD in 43,000 microscopy centers; about 61% of the analyses were conducted in the BRICS countries (Brazil, Russian Federation, India, China and South Africa) [37]. About 79% of the smears performed in the BRICS countries were used for initial diagnosis. On average, the unit cost for a smear was 1.77 USD, including materials, labor, and overhead expenses [35]. Several studies had shown that the accuracy of SSM improved when specimens were subjected to liquefaction, followed by the concentration of the *Mycobacteria* through overnight sedimentation or centrifugation [34,38,39,40,41,42]. However, the enhanced SSM performance provided by these pretreatment steps may not be sufficient to offset their increased cost, the complexity of their process, and potential biohazards.

Recent advances in bacteria preconcentration and the diagnosis of TB and multi-drug resistant tuberculosis (MDR-TB) include sophisticated techniques such as Xpert MTB/RIF, TB beads, liquid culture, centrifugation, filtration, and line probe assays [43,44,45,46,47]. However, these techniques are not necessarily accessible or affordable for those who need them the most [48]. Considering the high accuracy (~97%) and specificity (~99%) of the Xpert system relative to the culture standard [36], the World Health Organization issued a recommendation in 2010 to use Xpert MTB/RIF for the diagnosis of all persons with signs and symptoms of TB. However, the Xpert MTB/RIF assay entails a price of US$10 per cartridge. Thus, if this method was to be implemented for all people with presumed TB, the cost would exceed 80% of the total TB spending in low-income countries such as India, Bangladesh, Indonesia and Pakistan [49]. In 2014 and 2015, there were 33 and nine SSMs for every Xpert MTB/RIF test procured, respectively [50]. While high-end diagnostic methods are more accurate and/or specific than SSM, these techniques remain cost-prohibiting and inaccessible for people living in low-income countries where Mtb has a high prevalence. 

An essential aspect of TB is the substantial financial burden placed on patients and their families due to treatment and associated costs. For example, TB patients are often required to take absence leave from work, which, is unpaid in some cases, leading to a higher risk of financial struggle in the household [51]. Tanimura et al. reported the distribution of financial burden for the TB patient as 20% due to direct medical costs, 20% due to direct non-medical costs, and 60% due to income loss [52]. On average, the total cost was equivalent to 58% of reported annual individual income and 39% of reported household income [52].

In this context, accurate, rapid, and cost-effective diagnostic tests are paramount for reducing TB infection and its unacceptably high mortality rates, especially for an easily treatable disease [53]. The ambitious goal of the global “End TB Strategy” to diminish TB incidence by 90% and reduce TB mortality by 95% by the year 2035 is unlikely achievable without highly accurate yet low-cost tools to address epidemics in settings of poverty [54]. New tools must include improved point-of-care diagnostic tests that are delivered to low-income communities and at the first point-of-contact by patients in the healthcare system. Ideally, TB tests should be performed with the use of non-invasive sampling procedures, and results should be promptly delivered to the patients, allowing for a quick turnaround time for treatment in a single clinical encounter and hence avoiding the loss of patient follow up [54].

Thus, our strategy was to develop low-cost biosensing assay for rapid TB detection by employing modern advances in nanoparticle science and glyco-chemistry, thus resulting in an accuracy matching the performance of Xpert MTB/RIF [55,56] and standard culture. The nanoparticle-based colorimetric biosensing assay (NCBA) is based on the concept of the magnetically activated cell enrichment (MACE) technique using glycan-coated magnetic nanoparticles (GMNP). In this technique, the Mtb cells are isolated and enriched by applying a magnetic field to activate nanoparticle-bound Mtb cells without using any expensive antibodies or energy-consuming centrifuge instruments, thus eliminating the need for time-consuming growth of Mtb. The NCBA test involves the utilization of iron oxide nanoparticles with superparamagnetic properties. The incorporation of magnetic nanoparticles (MNPs) allows for significant improvements over other pre-concentration techniques due to their high surface-area-to-volume ratio and physicochemical properties. The MNP solution is colloidal in nature, providing stability, low sedimentation rates, and minimal precipitation due to gravitation forces. The MNPs are coated with glycan to facilitate their attachment to the bacterial cell wall through carbohydrate-binding protein sites, providing selectivity to the biosensing mechanism. There are three stages of specificity involved in this method: First, glycan–cell interaction is specific to the bacteria cell membrane through carbohydrate–protein binding. Second, the Ziehl–Neelsen staining used in the NCBA test is specific to acid-fast bacilli *Mycobacteria*. Third: the *Mycobacteria* present in sputum due to respiratory hemoptysis (i.e., intense coughing) is likely TB-causing bacteria.

The NCBA has been used to test sputum samples in Nepal (500 samples), Peru (1108 samples), and Mexico (24 samples) [55,56,57]. In the case of Nepal, all sputum samples were tested for TB by using three different methods: SSM, Xpert MTB/RIF, and the NCBA. In this study, SSM detected only 40% of the true-positive specimens, while Xpert and the NCBA successfully detected 100% of the true-positive samples. Neither one of the methods yielded false-positive results. Table 1 presents the results from the SSM (left panel) and the NCBA tests (right panel), using Xpert MTB/RIF as the standard for defining the number of true-positive and true-negative TB cases. Table 2 presents the performance characteristics for both SSM and the NCBA, including sensitivity, specificity positive predictive value (PPV), negative predictive value (NPV), and accuracy. As shown in Table 2, at a 95% confidence interval, SSM had a relatively low sensitivity of only 40% (29−52%), while the NCBA exhibited high sensitivity comparable to the Xpert system (95−100%). The accuracy of SSM was 90% (87–93%), while the accuracy of the NCBA was 100% (99–100%). Given the sample size and nature of the collected samples, the calculated prevalence for this cohort of patients was 16% (80 out of 500).

When samples were positive, the Xpert MTB/RIF system reported the bacterial load set by the manufacturer as very low, low, medium, and high. These four categories were used to estimate the equivalent load in SSM and the NCBA by matching the corresponding samples with Xpert results. Table 3 shows a comparison of the detection limit and dynamic range of the detection of the two techniques with respect to the Xpert system. As seen in the table, the NCBA yielded the same results as Xpert MTB/RIF at all levels of bacterial load. Conversely, SSM was unable to detect positive samples at the very low level and detected only 14% of true-positives at the low level, 48% at the medium level, and 79% at the high level. TB positive samples are normally distributed around the medium level, at which SSM exhibited a poor detection rate of less than 50%. 

The NCBA method significantly outperformed SSM with a lower detection limit for acid fast bacilli (AFB) of 10^2^ CFU/mL and a fast analysis time of 10–20 min. This diagnostic tool is facile (Figure 2), easily scalable, and inexpensive (0.10 USD/test). According to the Ministry of Health of Nepal, a low-cost TB diagnostic test with 70% accuracy could potentially save 300,000 lives just in Nepal over the next five years [58]. The NCBA technique shows promising potential for improving the TB control program in Nepal and other high-prevalence low-income countries. The deployment of the NCBA in remote rural areas would help increase case finding and case notification, thus supporting public health programs for fighting drug-resistant TB. There are nearly 600 microscopy centers distributed throughout Nepal in which the immediate implementation of the NCBA is possible. Similarly, this technique is applicable in many of the high TB-burden countries. In 2013, Desikan hypothesized that a universally accessible and rapid detection method with a sensitivity of 85% and specificity of 97% could save about 392,000 lives every year worldwide [33]. Thus, the developed NCBA technology may enable the “End TB Strategy” and lead towards a TB-free world.

### 2.2. Alerting Mercury Exposure in Artisanal Gold Mining Communities 

In South America, Africa, and Asia, millions of individuals are exposed to dangerous levels of mercury concentrations as a result of artisanal small-scale gold mining (ASGM) [59]. ASGM is a rudimentary gold mining approach that is performed by individuals or groups with little or no mechanization, often in informal (illegal) operational settings with toxic chemicals [60]. ASGM is composed of three main steps: crushing the ore into fines, mixing the fines with liquid mercury, and separating the mercury from gold by evaporating the mercury [61]. Often in unregulated occupational conditions, workers perform mercury evaporation by using open pits, which not only have severe adverse health effects for the workers that inhale the mercury vapor but also release the toxic vapor into the environment. ASGM recently exceeded combustion of coal as the leading anthropogenic source for mercury emissions globally [62]. The risk of exposure to mercury can lead to detrimental effects on the nervous, immune, reproductive, and digestive systems, induce infertility, reduce mental function, and induce kidney failure [63,64,65,66,67].

The global responsibility for reducing mercury emissions was recognized by the Minamata Convention in Switzerland in 2013. At the convention, over 140 countries signed a treaty committing to protect human health from mercury exposure [62]. The signatory countries pledged to “ban new mercury mines, phase-out existing mines, ensure the phase out and phase down of mercury use in a number of products and processes, develop control measures for emissions, and regulate the informal sector of ASGM” [62]. In order to mitigate mercury exposure and regulate mining operations, it is prudent for marginalized communities to monitor the presence of mercury in their water through low-cost, rapid, and facile devices. 

Several analytical methods have been developed for mercury determination in water. Standard laboratory techniques include cold vapor atomic absorption spectroscopy (CV-AAS) [68,69], cold vapor-atomic fluorescence spectrometry (CV-AFS) [70,71] and inductively coupled plasma mass spectrometry (ICP-MS) [72,73]. These spectroscopic techniques are highly sensitive and accurate but are often impractical for environmental applications due to the high cost of analysis. In addition, these standard methods require extensive user training, and the results often require days or even weeks to produce results, making them less suitable for rural communities [74,75,76]. Some field capable units are commercially available, namely based on direct mercury analysis (DMA) and handheld nanosensors/biosensors [77,78]. DMA is based on the principle of thermal decomposition (vaporization), followed by amalgamation and subsequent atomic absorption spectroscopy. While extremely accurate, DMA is cost prohibitive for low-income communities because commercial prices of US-manufactured equipment range between 13k and $30k USD. Perhaps inexpensive nanosensors/biosensors that are coupled with low-cost electrochemical techniques on portable devices are likely to be more suitable as tools for the on-site analysis of mercury, especially where ASGM is in practice. 

While there are many types of transduction methods for the low-cost determination of mercury, electrochemical methods are sensitive, quantitative, and may be the mechanism of choice for cost-effective rapid detection in the field [79]. The most common electrochemical method for ionic mercury detection is that of the anodic linear stripping voltammetry (ASV) techniques [74,80]. ASV is a two-step method of deposition/accumulation during the reduction of mercury ions and stripping during the oxidation of mercury ions along the surface of the electrode. As the mass transfer limit is reached in the reaction, the oxidative current forms a well-defined peak that can be used to calculate the concentration of mercury in the sample [81]. The efficiency of any electrochemical stripping test can be determined by calculating the percent change in oxidative current relative to baseline. 

Carbon-based nanomaterials are a popular choice for improving the electrochemical detection of mercury, as this type of material exhibits a high surface area, strong mechanical strength, excellent thermal conductivity, and high conductivity [82,83,84]. Some of the carbon nanomaterials in recent literature include glassy carbon [85,86], carbon nanotubes [87], graphene [88], and reduced graphene oxide [89]. While each of these nanocarbon materials is efficient for mercury detection via stripping voltammetry, some of the materials are complicated to fabricate and exhibit poor water solubility [90]. Among carbon nanomaterials, graphene and reduced graphene oxide (rGO) have the highest water solubility and one of the lowest fabrication costs. For these reasons, there is a growing trend to develop disposable, low-cost, graphene-based electrodes for field applications. 

Examples of low-cost graphene electrodes include screen-printed electrodes and conductive paper and plastic [74,91]. In 2014, Lin et al. (2014) [92] discovered a low-cost, one-step, conductive material when reducing graphene on a commercial polymer with a carbon dioxide infrared laser. Since then, multiple researchers have shown that laser scribing could be used to design electrodes to sense biomolecules by using infrared and ultra-violet light lasers [93,94,95,96]. While graphene is indeed a useful material in sensing, one of its problems is the tendency of graphene and graphene oxide to bind to a variety of materials in aqueous phase [97]. For this reason, sensor labs typically metallize graphene electrodes with a noble metal that has a specific interaction with mercury ions. These metals can be deposited by using simple electrodeposition methods or advanced techniques such as pulsed sono-electrodeposition [98]. Recently, Abdelbasir et al. 2018 [99] showed that copper nanoparticles recovered from waste cables can be used to detect ionic mercury by using linear sweep stripping voltammetry (LSSV). 

Low-cost, portable, mobile phone-based acquisition systems have been developed for mercury analysis in the field [100]. While this is significant for deploying sensors in low-income regions, the inexpensive-portable sensor-systems lack data analytics capability to transform the data into meaningful information that could be useful for the user. For example, the maximum concentration level for inorganic mercury in drinking water is 6 ppb [101]. However, bodyweight, ingestion rate, length of exposure, form and pathway of the contaminant, health of the individual, and concentration of mercury influences the degree of mercury toxicity [102,103,104]. Thus, a SNAPS tool may assist communities in acquiring data and extracting actionable information for decision support. 

Our group is currently working on developing the SNAPS platform for estimating the toxicity risk associated with the ingestion of mercury-contaminated water. This SNAPS platform is composed of a disposable graphene–nanocopper sensor that is coupled with a low-cost handheld potentiostat and a smartphone. The working mechanism of the platform starts with the detection of mercury present in the sample by using the graphene–nanocopper sensor. Next, selective electrochemical interactions between mercury and the electrode generate an electrical signal. The electrical signal is acquired and processed by the potentiostat to produce a current output. Then, computer software records the current output and transforms it into concentration data via calibration curves. Finally, a smartphone app is used by the user to enter the data for the following parameters: mercury concentration in water (from the sensor), bodyweight of the user, water ingestion rate, and length of exposure. Based on these parameters, the app runs an algorithm that includes a hazard quotient formula to generate an estimation of the risk of toxicity for the user [105,106,107,108].

We recently conducted a proof-of-concept demonstration of this SNAPS platform in a rural area that has been dramatically impacted by ASGM known as La Toma in Cauca, Colombia. Even though this SNAPS platform is in an early stage of development, it represents an example of how rural communities in developing countries may use sensors as a service to access data on mobile devices and extract actionable information to help make informed decisions. Figure 3 shows the progression of the proof-of-concept demonstration of the technology.

Mercury enters natural aquatic systems primarily due to the burning of mercury amalgam during the extraction of gold from raw ore.

## 3. Can We Overcome the Economic Barriers for Distributing Diagnostic Tools in Low-Income Settings?

Framing the issue of diagnostic tools in the context of technology leads us to recognize a vast spectrum. On one hand, ideas about telemedicine were proposed about 100 years ago [109], and on the other hand, milestones in computational speed occurred about 100 days ago [110]. It may be justifiable to suggest that technological barriers may not be the primary reason why many diagnostic tools are still absent from communities under economic constraint. The powerful incentive of lucrative profitability, in the short term, may not be realized by serving impoverished regions. 

Transaction cost [111] may be the over-arching factor that has multiple interpretations [112] but appears to be the economic barrier with respect to the reasons why accelerating the rate of diffusion of diagnostic tools in distressed communities continues to pose difficult challenges [113,114,115]. We must focus on value to the user or the extent of the benefit to the beneficiary’s environment and/or ecosystem (for example, the early diagnosis of tuberculosis in a patient may save the entire village from infection and epidemic). However, delivery of value is inextricably linked to cost, unless it is aimed to deliver philosophical or mythical messages [116].

In over-simplified terms, the convergence of the cost of the product and the cost to deliver the service contributes to transaction cost [117]. A plethora of costs and cost-incurring processes are involved, but we shall bypass the details. The physical product (in this case is the sensor) and the service is the solution delivery (SNAPS). Academics cannot control cost, but their contribution can impact implementation and use. A low-cost sensor from a lab must be manufactured, calibrated, evaluated, and sufficiently scaled if the outcome can still be claimed as a “low-cost” sensor that is capable of delivering value with respect to maintaining a certain pre-agreed quality of service (QoS) in keeping with the key performance indicators (KPI) that the users desire, demand, or deem necessary. 

In addition, a working sensor that is delivered to a user is useless without a visualization system to capture the data from the sensor. Stand-alone visualization devices (for example, blood glucose home monitors with dedicated devices to read the blood glucose strip and deliver data readout) add inordinate costs to the system. The alternative is to use a mobile phone as a platform to visualize the data from the sensor. The signal transduction from the sensor to the mobile phone calls for multiple layers of tools, technologies, and software (middleware), in addition to the functional use of a mobile phone. The presence of a mobile phone in any environment is contingent upon available cellular and/or wireless infrastructure to support its use. It may not be prudent to assume the presence of a telecommunications infrastructure despite the global penetration of such services [118,119,120,121]. Thus, even if a working sensor is at hand, the obvious process of signal to data transition and the visualization of the data involves multiple layers of capital expenses (infrastructure cost), as well as associated technologies and software.

Assuming that the above layers are in working order, the sensor data meets a “dead end” upon data visualization. A number (with units) is only meaningful if there is a relevant framework for interpreting such data, e.g., the combination of sensor data from mercury contamination expressed in terms of a hazard quotient score, which uses other vital pieces of information to assess health risk. It is the delivery of information based on sensor data that drives value. Taken together, the physical product is no longer the focal point of value. Information pertaining to the health of the user is the service that delivers value to the user. Transaction cost, therefore, is no longer a product-based entity; rather, it is the cost of service that must be feasible for the service to be delivered, disseminated, and adopted by a community.

Overcoming the economic barriers to deliver SNAPS will be virtually impossible if the chasm between product and service continues to overshadow the concept of value delivery to the user. The economic principle, which may work in impoverished nations, is rooted in micro-finance and micro-payments with low transaction costs [122,123]. The paradigm shift from “product sales” to delivery of “service” involves combining the product with resources (including retail mobile banking, infrastructure, telecommunications, cybersecurity, and customer service). Users pay only when they use the service. The latter lowers the transaction cost and hence the barrier to entry into vast markets of low-income users. It is not the product but the user experience that is the pivotal fulcrum for the inversion of traditional business models in the era of the Internet of Things (IoT) [124].

The PAPPU model was epitomized by the plain old telephone system (POTS), where the user paid only the “charge per call” which was reasonably affordable even if the per capita income was low. In this paper, we advocate for PAPPU as a metaphor for ethical profitability through social business models. In principle, the user may pay a penny for each use of a SNAP (suggested but not restricted to one penny). The “penny” is a placeholder for the financial design of an ultra low-cost nano-payment model, which, in the real world, may represent one Rupee (INR), one Yuan (CNY, RMB) or one Peso (COP). The PAPPU metaphor may evolve to become the generalized monetization mantra that signifies pay-a-price-per-unit wherever the principles of IoT may be deployed or embedded as a digital by design metaphor including ubiquitous sensing. The diffusion of connectivity may serve as a tool and IoT may be catalytic as a platform to better facilitate the practice of equality, equity and égalité. PAPPU offers an economic instrument for businesses to build a profit model based on economies of scale to serve low-income communities and abide by ethical profitability. PAPPU offers an alternative strategy for enterprises and businesses who are seeking to engage with the next billion users, albeit profitably, but within the realms of ethical profitability that can be sustained by the per capita income of these communities. 

The concomitant growth of infrastructure (e.g., affordable access to low latency, reduced jitter, high bandwidth wireless telecommunications, 5G, and trusted mobile banking) may be necessary to pave the road for the pursuit of PAPPU. The ability to escape the dead weight of old technology in the developing world may accelerate the implementation of PAPPU as an integral part of the socio-economic fabric of a product-less, service-based economy where payment per unit of service (one liter of municipal water, one kilo-watt hour of energy, or one gallon of sanitation waste) may become the new normal.

Implementing PAPPU may require alliances, public–private partnerships, or global consortia with an altruistic fervor to pay and pave for the synergistic integration that is necessary to promote SNAPS as services in low-income communities. The challenge is to bring to the table global organizations, benevolent individuals, and thoughtful governments who may choose to lead this effort to channel science to serve society for the less fortunate. We need new eyes, unbridled imagination, and the moral fabric of synergistic solutions that can wrap around—not to isolate—and protect, provide and promote acceptable solutions for remediable injustices.

## 4. Social and Ethical Considerations for the Development and Implementation of SNAPS

Social and ethical considerations are inextricably linked with the transformation of SNAPS from an academic vision to real-world implementations that may actually help people. Academics must remain cognizant of their ethical responsibility to discourage the misapplication and dissemination of misinformation about their inventions. In this section, we attempt to analyze some potential interactions between the social and technological domains, as well as how democratic approaches for technology creation and diffusion could favor the improvement of health outcomes for disadvantaged communities.

Since the introduction of the technology acceptance model (TAM) decades ago, several extended versions of this archetype have been proposed to elaborate a more comprehensive framework for predicting people’s intention to use a particular product or service [125,126,127]. The TAM and its variants have served as the guiding rationale behind R&D for a variety of commercial technologies that are mass-produced, including healthcare devices [128]. However, this model may be inadequate in the context of technology development for low-income communities [129]. It is worth noting that the ultimate goal of the TAM and related models is to forecast user behavior across a broad range of consumer populations, which means that the model focuses on highly generic predictors of technology acceptance. For instance, the TAM does not explicitly include any cultural or social variables, which is a significant limitation because social differences may contribute significantly to the variance in users’ attitudes towards technology [127,130]. However, the goal of SNAPS with the PAPPU concept is to provide an affordable sensor-analytics service platform to support decision-making and the enhancement of health outcomes for economically challenged groups. Thus, a useful model to guide the development of SNAPS should include bi-directional communication between researchers and users, and it should perhaps motivate researchers and users to change or adapt or better inform their behavior [131]. 

Trust in the technology [132] is quintessential for adoption and continued use, because technology is equally seen as a double-edged sword [133,134]. Driving positive impacts from the introduction of SNAPS in low-income regions may involve not only the transfer of fully functional technology but also the empowerment of the beneficiary communities by enabling the local mastery of the technology along with the possibility to re produce and even adapt the technology to local conditions. We believe this open-source approach to technology adoption is auspicious for supporting marginalized communities, especially when trying to avoid the known failures of the charitable approach of technology leapfrogging. For example, the WHO estimates that only 10%–30% of the medical devices that are donated to developing countries are used as intended; the remaining 70–90% end-up being dumped in landfills, thus contributing to more pollution problems and environmental health risks [135]. This situation is explained not only by the incompatibility of the technology with the locally available infrastructure but also to the lack of local capacity to adapt or fix the donated devices once they break [136]. Additionally, dependence on foreign technologies could lead to an imbalance of power in which the users have no option other than relying on the willingness of external entities to continue to deliver much-needed technology in their regions. Thus, if the goal is to make technology work effectively on behalf of society, we must divert from the mainstream handed-down from the top approach and enable society to create and transform technology in meaningful ways, in dispersed regions, and from the bottom-up.

Engaging the community through operational transparency may prevent public anxiety and may also facilitate the proper implementation of technology. Users’ understanding of the limitations and potential risks associated with SNAPS could be vital for setting clear expectations about SNAPS-assisted testing while avoiding misapplications of the technology. As Wallace et al. pointed out, the misuse of many direct-to-consumer screening tests could have caused an unnecessary increase in healthcare costs due to people’s overreaction to inaccurate readings from direct-to-consumer screening tests, as well as their subsequent demand for further testing with advanced clinical technology [132]. However, this concern is mostly relevant for developed countries in which people have access to healthcare systems where clinical testing is readily available for patients. In low-income settings, such as remote rural areas in developing countries, health care services are often dysfunctional or completely inaccessible. For marginalized communities, information from SNAPS could instead drive actions that are aimed at limiting the exposure to harmful biological vectors and chemical agents. Thus, communities in territories that suffer from prolonged government abandonment could greatly benefit from the democratic adoption of SNAPS to make informed decisions and solve their problems with more autonomy. Nonetheless, we agree that transparency and accountability from everyone involved in the process of technology deployment are paramount for protecting the users’ rights and integrity. 

## 5. Conclusions

Monitoring environmental contamination is essential to protect the public from diseases and other health issues. This monitoring requires accurate and cost-accessible sensor technologies to enable early warning capabilities for users to minimize negative impacts (Figure 4). The framework of SNAPS with PAPUU has the potential to pave the way for economically viable systems that can potentially be applied as tools to reduce local environmental risks and mitigate health problems that are derived from them. We envision that the use of SNAPS will increase low-income communities’ participation in the public/government planning process by providing data that they can use to fight for their right to public health care, clean water and adequate sanitation. By bridging smart technology with basic needs and public health, SNAPS will advance our understanding of how information can change public participation, having low-income communities’ representatives as ‘change agents’ that influence public policies and planning. These communities’ representatives benefit from rights-based arguments, evidence-based research, and effective data analyses. SNAPS have the potential to serve as an illustration of how empowering impoverished communities in their local context can strengthen democratic practice in their region. Grounded on an integrated perspective that takes social and ethical considerations into account, we foresee that SNAPS will shed some light to improve implementation of public health plans in underserved communities by increasing public participation in planning. Moreover, SNAPS could potentially become a new approach to achieve the United Nations Sustainable Development Goals 3 and 6: ensure healthy lives while promoting well-being at all ages and ensure access to water and sanitation for all, respectively. Furthermore, it could also help empower impoverished communities to obtain the rights they have been promised such as basic sanitation, clean water, and adequate health care services.

## Figures and Tables

**Figure 1 diagnostics-10-00022-f001:**
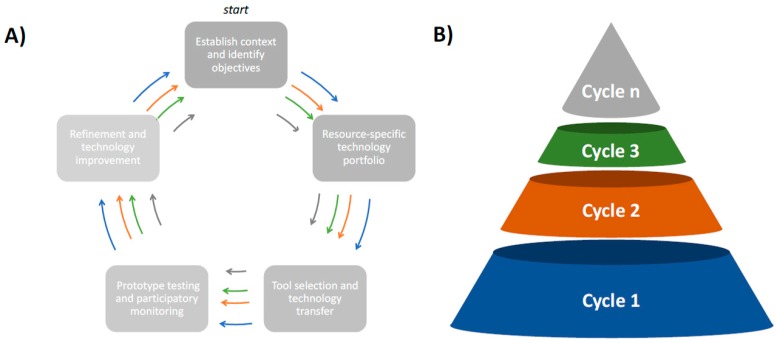
Overview of process in development of sensor analytic point solutions (SNAPS) for the examples shown below. (**A**) The process begins with establishing context, and each cycle concludes with technology refinement based on user feedback. The blue, orange, and green arrows indicate technology evolution by using established principles of circular feedback systems. (**B**) A conical representation of the blue, orange, and green cycles shown in Panel (**A**) indicate convergence toward a systems-level solution through feedback/refinement pathways. The total number of cycles is context-specific and proceeds from cycle 1 to cycle n.

**Figure 2 diagnostics-10-00022-f002:**
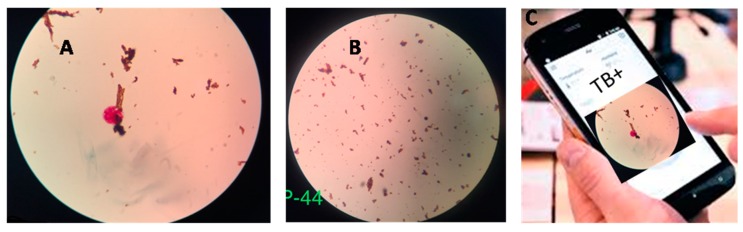
Typical nanoparticle-based colorimetric biosensing assay (NCBA) results for TB+ and TB− sputum samples, as viewed through the eyepiece of the bright field microscope. (**A**) The TB-positive sample (clumped red GMNP-AFB complex surrounded by brown GMNPs). (**B**) TB negative sample (dispersed brown GMNP). (**C**) Schematic of smartphone app for image processing and display of test results [55].

**Figure 3 diagnostics-10-00022-f003:**
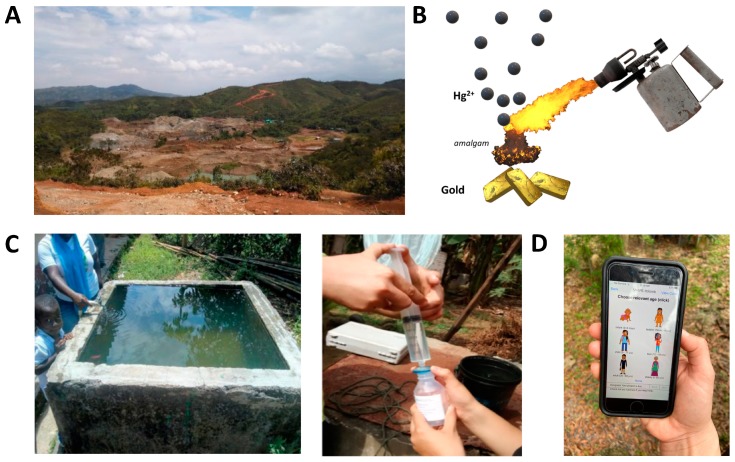
Demonstration of a SNAPS tool for assessing risk due to the inadvertent consumption of mercury in drinking water for gold mining communities in Colombia. The first step was to (**A**) characterize the local socioeconomic dynamics and (**B**) identify related routes of mercury exposure (in this case from smelting of amalgam). (**C**) Together with community members, we collected samples from local water sources. (**D**) These samples were tested with nanomaterial-enabled sensors. (**D**) Concentration data derived from sensors were transformed into customized information about the toxicity risk for specific user groups who were using a mobile app.

**Figure 4 diagnostics-10-00022-f004:**
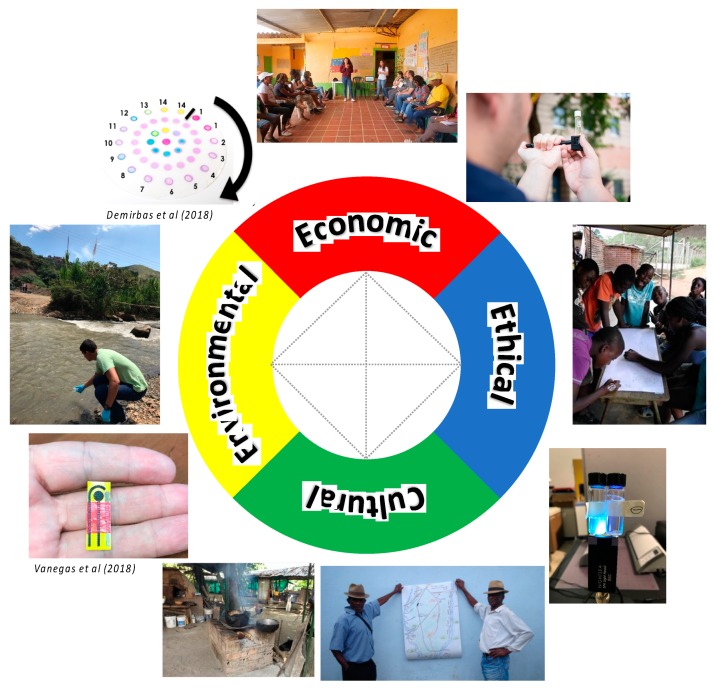
SNAPS converges with pay-a-penny-per-use (PAPPU) to establish a framework for sensor-as-a-service. The paradigm is rooted in economic, ethical, cultural, and environmental core values that synergistically act as a catalyst for the democratization of healthcare in underserved communities. Where noted, photos credited to Demirbas et al. [137] and Vanegas et al. [95].

**Table 1 diagnostics-10-00022-t001:** Results found by using Xpert MTB/RIF as the gold standard for true tuberculosis (TB) cases and non-TB cases [55].

SSM Test	True TB Cases	Non-TB Cases	NCBA Test	True TB Cases	Non-TB Cases
Positive test	32	0	Positive test	80	0
Negative test	48	420	Negative test	0	420

**Table 2 diagnostics-10-00022-t002:** Comparison of diagnostic performance [55].

Technique	Xpert MTB/RIF as the Gold Standard, % (95% CI)				
	Sensitivity	Specificity	PPV	NPV	Accuracy
SSM Test	40 (29–52)	100 (99–100)	100	90 (88–91)	90 (87–93)
NCBA Test	100 (95–100)	100 (99–100)	100	100	100 (99–100)

**Table 3 diagnostics-10-00022-t003:** Detection limit and dynamic range of detection of the two techniques with respect to the Xpert MTB/RIF categories [55].

Xpert MTB/RIF Categories **	Very Low	Low	Medium	High	Total
Xpert MTB/RIF	10	22	29	19	80
NCBA	10	22	29	19	80
SSM	0	3	14	15	32
% Detection (NCBA/Xpert)	100%	100%	100%	100%	
% Detection (SSM/Xpert)	0%	14%	48%	79%	

** The Xpert MTB/RIF assay provides semiquantitative readouts based on the cycle threshold (*C_t_*): very low = *C_t_* > *28,* low = *C_t_ 22–28*, medium = *C_t_ 16–22,* high = *C_t_* < *16.*

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
