# Peer review of "Sensor-as-a-Service: Convergence of Sensor Analytic Point Solutions (SNAPS) and Pay-A-Penny-Per-Use (PAPPU) Paradigm as a Catalyst for Democratization of Healthcare in Underserved Communities"

_diagnostics, 2020, doi:10.3390/diagnostics10010022_

Round 1

Reviewer 1 Report

The manuscript is clearly written, well structured, I recommend this paper for publication in Diagnostics.

Author Response

Thank you very much for taking the time to review our paper.

Reviewer 2 Report

This manuscript reviews efforts to develop practical sensor solutions for point-of-care diagnosis of infectious diseases or detection of environmental contamination and determination of its impacts on at-risk individuals utilising robust sensor hardware and easy-to-use software specifically tailored to underserved and disadvantaged communities. The authors commence their review by rightly pointing out the deficiencies of most reported sensor technologies that block their implementation in practical real world settings. They then go on to consider the key requirements for usable and affordable on-site sensor solutions using their suggested framework of sensor analytic point solutions (SNAPS). This is then clearly and logically supported by two major examples of such technologies in action and then conclude with more generalised commentary of the key challenges and barriers to implementation of sensor technologies in underserved communities encompassing both sensor technology and social science. The manuscript is clearly written, logically structured and easy to read. The focus of the review is appropriate given the urgent need to address sensor technology to improving health and well-being in communities where healthcare delivery is highly challenging. The authors have chosen to focus their examples mostly on two key technology case studies which does tend to narrow the technology scope of the review. However, I feel that this approach is justified as these examples bring to the fore some of the key issues faced by SNAPS-type technologies and the authors have followed this up with more generalised discussion sections. There are some minor points for the authors to consider and these are detailed below.

Figure 1. You have included a set of three coloured arrows in this diagram and mentioned them in the figure legend but it is not clear what they are supposed to represent, e.g. parallelism in the SNAPS development process, as opposed to using a single flow arrow in each stage of the process. Consider clarifying this aspect of the diagram. Page 5 lines 224-226 "The MNPs are . . . the biosensing mechanism." I think you should be a bit clearer about how specificity is conferred in this assay. Simply mentioning carbohydrate-binding protein sites does not suggest specificity to tuberculosis bacteria on and above other bacterial species. Some more detail is needed here. Page 6 lines 238-240 "Probably because sputum . . . close to 100%." This statement only really applies to the SSM method. This statement is not central to the main point here, which is that the SSM method suffers from high false negative rates, i.e. suffers from quite poor sensitivity. Table 3. So that the data in this table better suggests actual quantified detection limits, you should include mention of the concentrations or concentration ranges (CFU/mL) for each of the categories, perhaps as a footnote to the table. Page 12 line 520 ". . . protecting the users' rights and integrity." It is not clear to me how you are proposing that diagnostic technology deployment would protect the users' integrity. Are you meaning that they would be empowered to complete an initial self-diagnosis without need for initial outside intervention? Clarify your meaning at this point in the manuscript. Figure S1. You present this figure based on a simple Boolean search but later you go on to complete a manual screen to restrict findings to those actually directly related to the topic. Really, this pre-screen should also have been used on the results presented in Figure S1. Figure S2. I think this figure could be deleted as it does not support any key conclusions of the review and does not add much useful information.

Author Response

Dear reviewer,

We are grateful for the valuable feedback you provided for our manuscript “Sensor-as-a-service: Convergence of Sensor Analytic Point Solutions (SNAPS) and Pay-A-Penny-Per-Use (PAPPU) Paradigm as a Catalyst for Democratization of Healthcare in Underserved Communities”.

We conducted a thorough revision of the document and addressed each of your comments as follows (all modifications are highlighted in red within the body of the manuscript):

Comment: Figure 1. You have included a set of three coloured arrows in this diagram and mentioned them in the figure legend but it is not clear what they are supposed to represent, e.g. parallelism in the SNAPS development process, as opposed to using a single flow arrow in each stage of the process. Consider clarifying this aspect of the diagram.

Answer: This is a great point. We have improved the diagram and included concepts for both parallel and single pathway development.

Comment: Page 5 lines 224-226 "The MNPs are . . . the biosensing mechanism." I think you should be a bit clearer about how specificity is conferred in this assay. Simply mentioning carbohydrate-binding protein sites does not suggest specificity to tuberculosis bacteria on and above other bacterial species. Some more detail is needed here.

Answer: We appreciate the opportunity to clarify this point. Indeed, there are three stages of specificity in the SSM method: First, glycan-cell interaction is specific to bacterial surface through carbohydrate-protein binding. Second, the Ziehl-Neelsen staining used in SSM is specific to acid-fast bacilli Mycobacteria. Third: Mycobacteria in sputum due to respiratory hemoptysis (i.e., intense coughing) is likely TB-causing Mycobacteria (such as M. tuberculosis). We included this information in lines xx-xx.

Comment: Page 6 lines 238-240 "Probably because sputum . . . close to 100%." This statement only really applies to the SSM method. This statement is not central to the main point here, which is that the SSM method suffers from high false negative rates, i.e. suffers from quite poor sensitivity.

Answer: We agree that this is an unnecessary and possibly distracting statement, therefore we removed it from the manuscript.

Comment: Table 3. So that the data in this table better suggests actual quantified detection limits, you should include mention of the concentrations or concentration ranges (CFU/mL) for each of the categories, perhaps as a footnote to the table.

Answer: This is a very good point. In order to compare the performance of the NCBA method with other TB detection techniques, we used the same classification ranges set by the manufacturer of the Xpert MTB/RIF system as the reference standard. In fact, when the tested sample is positive for TB, the Xpert MTB/RIF assay provides semiquantitative results based on the bacterial load associated with one of four possible cycle thresholds (Ct): very lowº Ct>28, lowº Ct 22-28, mediumº  Ct 16-22, highº  Ct<16. We included this information as a footnote in Table 3.

Comment: Page 12 line 520 ". . . protecting the users' rights and integrity." It is not clear to me how you are proposing that diagnostic technology deployment would protect the users' integrity. Are you meaning that they would be empowered to complete an initial self-diagnosis without need for initial outside intervention? Clarify your meaning at this point in the manuscript. Answer: Thank you for this question. The last statement ". . . protecting the users' rights and integrity." Was written in reference to the concerns raised by Wallace et al (2013), who argued about the problem created by some companies in the U.S. that are commercializing home screening tests using misleading marketing strategies. Because the aim of these companies is to profit from sales (not necessarily to improve health outcomes of communities), there is a lack of transparency about how the technology works and what are the technical limitations (e.g. this type of information may be protected by patent rights); in those cases, users have unclear expectations about the screening tests, they may or may not use the technology as intended, and they may end-up taking detrimental decisions for themselves based on inaccurate/misinterpreted results from these screening tests. 

The intention of our last statement is to advocate for public disclosure of information about the technology as part of the deployment process, including details about the recommended operation, benefits and limitations of the testing, and potential risks and harms. We made a subtle modification statement to clarify this point.

Comment: Figure S1. You present this figure based on a simple Boolean search but later you go on to complete a manual screen to restrict findings to those actually directly related to the topic. Really, this pre-screen should also have been used on the results presented in Figure S1. Figure S2. I think this figure could be deleted as it does not support any key conclusions of the review and does not add much useful information.

Answer: We appreciate your feedback on the supplemental section. We removed Figures S1 and S2 since these figures didn’t support our discussion points.

We hope these answers satisfy your questions and suggestions. We believe that these modifications were very important for strengthening our paper. 
